# Effect of Apartment Community Garden Program on Sense of Community and Stress

**DOI:** 10.3390/ijerph19020708

**Published:** 2022-01-09

**Authors:** Sang-Mi Lee, Hyun-Jin Jang, Hyung-Kwon Yun, Young-Bin Jung, In-Kyoung Hong

**Affiliations:** Urban Agricultural Research Division, National Institute of Horticultural and Herbal Science, Rural Development Administration, 100 Nongsaengmyeong-ro, Iseo-myeon, Wanju-gun 55365, Jeollabuk-do, Korea; jin23002300@naver.com (H.-J.J.); yun0309@korea.kr (H.-K.Y.); jybin77@korea.kr (Y.-B.J.); inkyoung63@korea.kr (I.-K.H.)

**Keywords:** apartment housing, intervention, community awareness, autonomic nervous activity, horticultural therapy

## Abstract

Apartment housing has become a dominant form of urban residence. High dwelling density in apartment complexes causes frequent conflicts and disputes. To counter this, it is necessary to promote a sense of community among residents with programs such as a customized horticultural program for the introduction of a community garden in an apartment complex. This study was conducted to investigate the effect of a community garden program in an apartment complex in fostering residents’ sense of community and reducing stress. Experiments were performed in three groups: a group participating in the program based on the sense of community theory (SCG; *n* = 11), a group participating with a focus on horticultural education (HEG; *n* = 11), and a non-participation group (NPG; *n* = 10). The experimental results revealed that the sense of community was significantly higher in the SCG than in the HEG and NPG. The results suggest that the SCG positively affected the sense of community, overall energy, ratio between sympathetic and parasympathetic nervous systems, and stress resistance. Considering these results, community garden programs with appropriate interventions to promote a sense of community are more effective in improving community life and reducing stress than programs based on horticultural education.

## 1. Introduction

### 1.1. The Purpose of the Study

According to housing statistics in Korea, the proportion of apartment housing has gradually increased from 58.4% (8,576,013 units) in 2010 to 62.3% (11,287,048 units) in 2019 [1]. Hence, it has become a dominant urban residential form. While the quality of dwelling conditions has improved, the independence of housing units, which is the greatest advantage and disadvantage of apartment housing [2], and the high dwelling density, frequently lead to conflicts and disputes among residents [3]. These conflicts are related to parking, inter-floor noise disturbance, second-hand smoking, pets, and the installation of electric vehicle charging stations. Hence, intensifying individualism, indifference toward neighbors, and lack of communication result in a reduced quality of life and increased conflicts, thus emerging as a new social problem. Moreover, apartment building designs mainly reflect suppliers’ interests and present stimuli to which residents must adapt. Such residential environments increase their stress levels [4].

To resolve conflicts between residents and reduce their stress, they should have a practical interest in sublimating neighborhood conflict by building a sense of community and developing community values [5]. Hence, programs should promote mutual communication through a sense of community, along with appropriate interventions to alleviate the conflict and stress associated with residential environments.

An apartment community garden is a place where residents can freely exchange opinions and information, providing an arena for social and cultural sharing [6]. Community members can easily participate in horticultural activities in a non-discriminatory and non-threatening natural environment [7], which helps achieve psychological stability, including nonverbal exchange and stress reduction [8]. Furthermore, horticultural plants inspire emotional motivation and help the participants feel confident and responsible while growing plants, helps gain a sense of accomplishment [7], and fosters emotion management strategies and skills by promoting self-control and alleviating negative emotions. This harnesses their minds with a mechanism that enables the positive adjustment of emotional stimuli caused by stressors [9]. Horticultural activities have physical and mental therapeutic effects by integrating sensation and perception while performing physical functions that involve all body parts based on basic sensorimotor experiences [10]. Therefore, a customized horticultural program using an apartment community garden can help increase residents’ sense of community and reduce stress.

Against this background, this study was conducted to verify an apartment community garden program’s effect on building a sense of community and reducing stress among residents.

### 1.2. Literature Review

#### 1.2.1. Community Conflict

Due to the introduction and expansion of apartments in Korea, large-scale apartment construction projects were carried out as part of the national industrialization policy and the increase of the urban population. The housing style of living in apartments, which has grown rapidly, has formed an independent and closed culture due to its unique structure and characteristics, revealing disconnection and various conflicts among its members and raising social concerns [11]. Due to the nature of community housing, apartment houses can cause several problems as many people live together in the same building, which explains the inevitability of conflicts between residents [12].

Among the problems that can be seen as the scope of conflict are daily ones between residents, conflicts between residents and management staff, and civil disputes over apartment management and operation [11]. Among them, the factors and details of the conflict can be classified into problems such as noise between floors, smoking, parking, pets, and basic order violations [11], as well as drinking disturbances, food waste, throwing trash from higher floors, leakage between floors, and pipe blockages, which are considered major problems which are difficult to solve [5]. Among them, in the case of noise disturbances between floors, there are many cases in which the conflict persists and intensifies, leading to crime or legal disputes [11].

#### 1.2.2. A sense of Community

McMillan & Chavis (1986), who had a great influence on the study of community spirit, defined community spirit as “a common belief that can satisfy the needs of members through the feeling of belonging, important feelings for each other and groups, and commitment to stay together.” They proposed four components of community spirit: membership, mutual influence, integration and fulfillment of needs, and shared emotional connection [13]. It is said that the role of a community is to provide a place for an individual and to surround their territory with fences, and the community is formed through public efforts [14]. Therefore, public efforts are needed to form a community, and a study by Han (2021) found that the higher the sense of community among residents, the higher the community satisfaction. Community participation and community satisfaction showed a positive relationship.

To create a desirable community in apartments, the purpose of this study, hardware must be provided as a physical environment, such as facilities for residents to communicate and exchange their thoughts. In addition, software such as program operation activities for residents to participate in is needed, and through this, human ware should be harmonized, which is inspired by community spirit.

#### 1.2.3. Community Garden

The community garden can be considered as an expanded kitchen garden with a social role [15]. Community gardens must be noted for their role in meeting social needs compared to general parks, and the qualitative value of these community gardens is found along with the following: open spaces for citizens, space for urban agriculture, community values through shared assets, garden education and training programs, and city sustainability [16].

Community activities centered on community gardens provide various benefits to participants. Charles Lewis said, “People desperately feel the value of plants by seeing them grow according to the care they receive and how they reveal the pain of patience and are not narrow-minded.” According to his book, “community gardens are an environment of recovery for the stress and social loneliness of urban life without vitality.” In addition to the personal benefits that can be obtained through community activities as above, people also experience social benefits [15]. For reference, in Park Se-young’s study [17], the creation of community gardens and the expansion of private garden sharing ranked third in terms of importance and first in terms of satisfaction as a result of the importance-satisfaction analysis of the basic plan for garden promotion.

## 2. Methods

### 2.1. Study Design

This study used quasi-experimental data of non-equivalence control group design. A quasi-experimental study design based on pre–post observations was used to verify an apartment community garden program’s effect on building a sense of community and reducing stress among residents of apartment buildings, using three groups: (1) group participating in the apartment community garden program based on the sense of community theory (SCG), (2) group participating in the apartment community garden program with a focus on horticultural education (HEG), and (3) non-participation group (NPG), as outlined in Table 1. 

All three groups underwent preliminary tests on community revitalization and stress before the program was implemented. After the preliminary investigation, (1) the SCG Group participated in the apartment community garden program based on the sense of community, (2) the HEG Group participated in the apartment community garden program focusing on horticultural education, and (3) the NPG Group did not participate in the program after the pre-test. After the program was completed, a post-test was conducted with the same title as the pre-test on all three groups. 

### 2.2. Participants

There were no inclusion or exclusion criteria for participation other than being apartment residents. The participants were recruited voluntarily. G* Power Version 3.1.9.6 (Franz Faul, Universität Kiel, Kiel, Germany) was used to determine the sample size. Subsequently, it determined the appropriate number of samples was 47, and the total number of recruits was 70; however, the number of samples used for the final analysis was 32, considering the program participation rate and the fidelity of the questionnaire responses. Thus, a total of 32 residents from three apartment complexes in three different cities in Korea participated in this study, resulting in two experimental groups and one control group: SCG consisting of 11 residents from the I apartment complex located in J city who had participated in a related program, HEG consisting of 11 residents from the J apartment complex located in S city who had participated in a related program, and NPG consisting of 10 residents from the L apartment complex located in J city with no prior experience with similar programs.

The participants were recruited on a voluntary basis through respective apartment management offices and recruitment posts. The researcher provided detailed explanations of the study and obtained signed consent regarding participation and the use of personal information. The study was approved by the Institutional Review Board of Dankook University (IRB approval number: Dankook University 2019-09-004). Figure 1 presents the CONSORT (Consolidated Standards of Reporting Trials) flow diagram for individual controlled trials.

The mean age (± standard deviation) of the participants was 60.7 ± 10.3 years (Table 2), and the sample comprised four males and 28 females (Table 2). No statistically significant age or sex differences were observed among the three groups.

### 2.3. The SCG Program

#### 2.3.1. Program Implementation Site

The SCG program was implemented in the community garden within the I apartment located in J city. There are 44 garden beds measuring approximately 4 m^2^ in empty spaces around the management office and the entrances of some buildings. Pergolas, chairs, tables, irrigation systems, and gardening tool storage boxes were provided, and the paths were paved with bricks (Figure 2). Apartment community facilities (the library and senior citizen hall) were used for indoor activities.

#### 2.3.2. Program Characteristics

The SCG program was based on the sense of community theory developed by McMillan and Chavis [18]. They proposed four elements that define a sense of community: membership, influence, integration and fulfillment of needs, and shared emotional connection.

The first element, “membership,” has five attributes: a common symbol system, boundaries, emotional safety, personal investment, sense of belonging, and identification, which interact among themselves and influence the awareness of people in and around a community [19,20]. To enhance membership identity, the program used a common symbol system of uniform caps, gloves, and scarves and set boundaries between the participants and outsiders by providing each participant with a community garden tag representing the geographic features of the apartment complex. The unique value of shared plant cultivation-related knowledge, which distinguishes the participants in the SCG program from those engaged in simple community garden cultivation, and the aforementioned common symbol system, were used as boundaries and also provided the members with emotional safety. Boundaries, which can be a geographical indication or similar interest, help identify members [18], and the common symbol system clarifies the boundaries of the community. In a community with clear boundaries, members experience emotional stability by sharing common interests.

In addition, personal investments for the community were encouraged by installing baskets to share the harvests and maintaining tool storage boxes and apartment community flower beds. Program experts provided guidance and encouragement, so that emotional safety and personal investment could lead to a sense of belonging and identification.

To empower the participants with the second element, “influence,” rule-setting, team formation, and role distribution were performed; influence was expanded to the entire apartment complex by sharing baskets with message cards or handmade articles using the harvest (e.g., flower arrangements, pressed flower bookmarks, herbal soaps, marigold teas, and marigold-dyed scarves) with other residents. Those who received community garden gifts attempted non-face-to-face communication, such as leaving a thank-you message in the sharing basket. Moreover, children from the daycare centers within the complex sowed autumn vegetables, such as cabbages and radishes, together.

Regarding the third element, “integration and fulfillment of needs,” the convergence of participants’ opinions about plants and the activities before the program, was reflected in the planning. Their needs for not only materials, such as compost, seeds, seedlings, harvests, and gardening tools, but also for knowledge of plant cultivation and use of the gardening products were considered and reflected in activity preparations.

Activities undertaken to induce “shared emotional connection,” the fourth element, involved encouraging a sense of sharing; this not only included sharing the space of the apartment community garden, but also time spent together in the program and daily activities, and joint efforts to overcome crises such as pest infestation. This element also comprised undertaking field trips to nearby locations, sharing the gardening experiences with other members, and tending to the garden beds.

The applications of the sense of community theory to the apartment community garden program are outlined in Table 3.

The SCG program (Figure 3) consisted of 19 sessions held every alternate month (March to November 2019). The contents of each session were as follows: Session 1: orientation; Session 2: uniform cap production and rule-setting; Session 3: gardening tool preparation, role distribution, and soil preparation; Session 4: harmonious planting plan; Session 5: understanding companion plants, Session 6: gardening crisis (pest management); Session 7: building earthworm compost boxes and brunch making and sharing; Session 8: herbal soap making and sharing; Session 9: flower arrangement and sharing; Session 10: making plates and bookmarks decorated with pressed flowers; Session 11: making tomato and plum pickles; Session 12: sowing autumn seeds; Session 13: sowing autumn plants with the children in the daycare centers in the complex; Session 14: making marigold tea; Session 15: transplanting Korean cabbage seedlings; Session 16: field trip to the National Institute of Horticultural and Herbal Science; Session 17: dyeing handkerchiefs with marigold; Session 18: planting daffodil bulbs in the apartment community flower bed; Session 19: garden party (Table 4). The SCG program was led by an urban agricultural manager and an assistant. They conducted the program after receiving training on the sense of community intervention from the program developer and thoroughly understanding how to administer the intervention program.

### 2.4. The HEG Program

#### 2.4.1. Program Implementation Site

The HEG program was implemented in the community garden of the J apartment complex located in S city, Korea. There are 30 garden beds each measuring 5 m^2^ on the edge of the complex, adjacent to the apartment management office. Chairs, irrigation systems, and gardening tool storage boxes were provided, and the paths between the garden beds were covered with grass (Figure 4). The community welfare center, an apartment community facility, was used for indoor activities.

#### 2.4.2. Program

The HEG program (Figure 5), which did not involve intervention for community building, was implemented in 19 sessions held every Friday (2–3 times a month) for 11 months (March to November 2019). The contents of each session were as follows: Session 1: orientation; Session 2: setting rules, voting to select a leader, creating a common garden, and planting leafy vegetables; Session 3: horticultural education focusing on nutritional management and planting flowers in the flower beds around the community garden; Session 4: understanding the utilization and characteristics of herb cultivation and planting flowers in the flower beds around the community garden; Session 5: horticultural education focused on pest management and planting seedlings of fruit vegetables (pepper and eggplant); Session 6: soil education and prop construction; Session 7: planting succulent plants; Session 8: small-unit compost packaging; Session 9: potato harvesting; Session 10: making a plate garden; Session 11: pest control and field cleanup; Session 12: community field preparation for the cultivation of autumn plants; Session 13: kimchi vegetable planting and individual plant management; Session 14: handkerchief dyeing with marigold; Session 15: field trip to the forest on the outskirts; Session 16: garden pest management and organic liquid fertilizer preparation; Session 17: understanding herbal farming characteristics and herbal beverage preparation; Session 18: autumn plant cultivation education and compost application for autumn plants; Session 19: flower-pot production (Table 5). The program was led by an urban agricultural manager and an assistant. The researcher did not mention the intervention for the sense of community to them and asked the participants to focus on learning how to grow plants.

### 2.5. Measurement Tools

The effect of the SCG program aimed at fostering a sense of community was verified by a pre–post survey analysis using questionnaires and biomarkers. The following were measured: sense of community, perceived stress, and measured stress. 

#### 2.5.1. Sense of Community

For the pre- and post-intervention measurements of the sense of community, the scale developed by McMillan and Chavis [18] and used in the studies by Park [21] and Choi and Jeong [22], was employed. The items were modified by Jeong [23]. The scale consists of four sub-dimensions: membership (three items), influence (three items), fulfillment of need (three items), and emotional connection (three items). Each item is rated on a 5-point Likert scale (1 = strongly disagree, 2 = disagree, 3 = neutral, 4 = agree, and 5 = strongly agree). The higher the score, the higher the sense of community. Cronbach’s α value in the previous study was 0.754 for membership, 0.678 for influence, 0.763 for the fulfillment of needs, and 0.817 for emotional connection [23]. In this study, they were 0.655 for membership, 0.810 for influence, 0.903 for the fulfillment of needs, and 0.876 for emotional connection.

#### 2.5.2. Stress

For stress measurements, the stress scale and biomarkers were used.

Perceived stress: The Psychosocial Well-being Index Short Form (PWI-SF) was used to measure stress levels. The PWI-SF is an abridged version of the 45-item Psychological Well-being Index (PWI) scale based on the GHQ-60 developed by Goldberg [24] and revised as an 18-item scale in two stages to adapt it to the Korean situation. Each item of the PWI-SF is rated on a 4-point Likert scale (1 = never, 2 = sometimes, 3 = mostly, 4 = always), where a higher score indicates a higher level of psychosocial stress. Cronbach’s α for the PWI-SF in the original study was 0.926 and, in this study, 0.870.

Measured stress (biomarkers): As biomarkers for stress level measurements, several autonomic nervous activities were measured using uBioMacpa (BioSense Creative Inc., Seoul, Korea), a test device whose safety and reliability are approved by the Korea Food and Drug Administration (KFDA). It can effortlessly measure pulse waves using a non-invasive method and heart rate variability (HRV) through a detailed analysis of minute changes in heart rate by sensing the changes in light reflection of capillaries at the fingertips.

Since stress levels can be determined by synthesizing the values of each item, such as pulse diversity values, heart rate tables, and autonomic nerve balance, the autonomic nervous system response based on HRV is measured to observe stress levels. Seven indicators were analyzed: total power (TP); low frequency (LF); sympathetic nerve activity in the low-frequency region; high frequency (HF); parasympathetic activity in the high-frequency region; the LF/HF ratio, representing the balance between the sympathetic and parasympathetic nervous systems; the standard deviation of the NN interval (SDNN), which is an index reflecting the physiological resilience against stress; and the root mean square of standard deviation (RMSSD), which indicates the heart stability and the mean beats per minute (BPM). These data were analyzed using uBioMacpa Pro Version 1.01 (BioSense Creative Inc., Seoul, Korea).

TP is the total power value including both LF and HF, that is, the overall activity of the autonomic nervous system. The sympathoactivation (LF) is high mainly in a tense or excited state, and parasympathetic nerve activity (HF) is high in a sufficiently rested or relaxed state. The mean deviation (RMSD) is also expressed as cardiac stability to check the degree of parasympathetic activity and is low in anger, anxiety, and fear [25].

### 2.6. Statistical Analysis

The final sample used for the analysis of this study was 32, which was less than the 47 samples derived from G*power Version 3.1.9.6 (Franz Faul, Universität Kiel, Kiel, Germany).

Hence, a nonparametric test was performed for statistical analysis before program implementation, and the Kruskal–Wallis test was performed on the data collected to test the homogeneity between groups using the SPSS Win.23.0 program. To test the between-group effects on the post-intervention values, the Kruskal–Wallis test and Bonferroni correction post hoc test were performed for the homogeneous items in the pre-test. The baseline-adjusted analysis of covariance (ANCOVA) and Bonferroni correction were performed as post hoc tests on the non-homogeneous items. For within-group pre–post comparisons, the Wilcoxon signed-rank test was used.

## 3. Results

### 3.1. Pre Homogeneity Test 

Table 6 outlines the pre-homogeneity of the sense of community in the SCG, HEG, and NPG. No significant differences were observed in the overall level of sense of community (*p* = 0.048) and three of its four sub-dimensions: membership (*p* = 0.413), influence (*p* = 0.652), fulfillment of needs (*p* = 0.183), and emotional connection (*p* = 0.563). Thus, the pre-homogeneity of the three groups was established. However, intergroup differences were observed in the sense of community (overall) (*p* = 0.048), and hence, ANCOVA was used to analyze the post-intervention values. Table 6 presents the results of the pre-homogeneity stress test between the three groups. No significant intergroup differences were observed in TP, LF, SDNN, and mean BPM, which establishes pre-intervention homogeneity. However, in perceived stress (*p* = 0.005), the values of HF (*p* = 0.008), LF/HF (*p* = 0.011), and RMSSD (*p* = 0.005) were not homogeneous among the three groups, which were then subjected to ANCOVA to analyze the post-intervention values.

### 3.2. Post-Intervention Intergroup Comparison

A post-intervention comparison of the sense of community scores between the SCG, HEG, and NPG revealed that the overall sense of community score was statistically significantly higher in the SCG than in the HEG and NPG. The SCG scored statistically significantly higher than the HEG and NPG in the sub-dimension of sense of community, namely, emotional connection (Table 7). Table 7 presents the results of the post-intervention comparison of the mean stress scores between the three groups. No significant intergroup differences were observed in the perceived stress levels, LF/HF, and RMSSD. The TP, LF, and SDNN values were statistically significantly higher in the SCG than in the HEG and NPG.

### 3.3. Within Group Pre-Post Mean Comparisons

Table 8 outlines the results of the within-group pre–post mean comparisons. The SCG showed an increase in the mean values of all four sub-dimensions of sense of community but without reaching statistical significance; in contrast, the overall score increased statistically significantly (*p* = 0.028). Within the HEG, the post-intervention mean decreased in all sub-dimensions except membership, but without statistical significance. Within the NPG, the post-intervention mean increased in all sub-dimensions of sense of community. Table 8 presents the results of the within-group pre–post comparisons of mean stress scores. Within the SCG: TP, LF, and HF increased with statistical significance, and TP, LF, and SDNN decreased significantly within the HEG. The NPG showed no significant changes in the pre–post comparisons.

## 4. Discussion

### 4.1. Sense of Community

In this study, SCG had statistically significantly higher scores in the sense of community than the HEG and NPG. The SCG scored statistically significantly higher than the HEG and NPG in the sub-dimension of sense of community, namely, emotional connection (Table 7).

Considering the findings of a previous study by Jeong [16], which showed that fulfillment of needs has a significant effect on the social activities of urban community garden users and emotional connection has a significant effect on social activities and user satisfaction, the SCG’s high scores regarding these two aspects can be attributed to the members’ active social exchanges within and outside the community. For example, the members had external communication and exchange through activities such as investment and sharing of personal resources in the form of sharing of baskets and time, and knowledge sharing in the form of activities with the children of the daycare centers within the apartment complex. Moreover, it is ascribable to a high level of satisfaction with the utilization of the community gardens via the fulfillment of needs through the support received, such as gardening tool storage boxes, plant seedlings and seeds, and horticultural knowledge.

Although not statistically significant, the sense of community of the NPG, which did not participate in any program, tended to increase, whereas that of the HEG, which participated in the program with a focus on horticultural education, tended to decrease. This could be because the program fosters communication and exchange among participants through constant face-to-face interaction, active information sharing, and voluntary activity requests [26]. This result differs from Choi’s [27] study, which showed that communication with other households increased through gardening activities in the community garden and that the garden serves as a community sphere within the apartment complex. However, it is consistent with Park’s [28] findings that the sense of community is higher in community gardens associated with various programs and activities than in those that only provide horticultural education.

These findings highlight the importance of setting a clear purpose, be it community building or food production [28], and including appropriate intervention programs for all residents to boost a sense of community, along with horticultural education, when operating a community garden program in an apartment complex where active community life is important. In this context, community garden specialists who can guide and develop such interventions are urgently needed. In response to the growing interest in leisure-oriented urban horticulture centered on community gardens, the Korean government is offering a national professional certification system called “urban agricultural manager.” The findings of this study suggest that urban agricultural managers should be equipped with the ability to provide interventions to foster community building, along with the knowledge and skills of farming and gardening.

### 4.2. Stress

No significant changes were observed in the between-group post-intervention means and within-group pre–post values of perceived stress. However, the SCG and NPG had statistically significantly higher TP, LF, HF, and SDNN values than the HEG, which had a significantly higher mean BPM than the NPG (Table 7). The within-group pre–post mean comparisons of stress showed that TP, LF, and HF significantly increased in the SCG and that TP, LF, and SDNN significantly decreased in the HEG (Table 8).

This suggests that an apartment community garden program based on a sense of community has a positive effect on the overall energy (expressed as TP) and the sympathetic and parasympathetic nerve activities. TP is a measure of overall energy reflecting the general control capacity of the autonomic nervous system (ANS), and an increase in its points leads to an increase in the overall energy level and improvement of ANS activity [29]. A TP-related study [30] reported that the exercise group had a higher TP value than the non-exercise group. Similarly, a study on the effect of regular yoga practice on middle-aged women’s ANS activity [31] reported an eight-week yoga intervention significantly increased their mean TP value, boosting overall ANS activity.

LF, which is related to sympathetic nerve activity, increases due to tension and anxiety, and decreases due to fatigue, thereby depleting bioenergy [32]. HF indicates parasympathetic nerve activity and is high in a state of sufficient rest and low when experiencing stress, anger, worry, anxiety, and fear. A lower HF in physically healthy people indicates a higher level of perceived stress [33]. From these previous findings, it can be inferred that the SCG program has a positive effect on the ANS response, such as overall energy level, rest, and stress reduction. 

Furthermore, the SCG showed higher stress resilience, as expressed by SDNN (Table 7), which is a stress index indicating the adaptability of the body against the stress caused by external factors. Hence, as an index reflecting the physiological resilience against stress, a lower SDNN indicates reduced stress resistance, which means reduced stress-coping ability and impaired control ability of the ANS as well as overall health status [34,35]. From this data, it can be concluded that the SCG program has a positive effect on the stress coping ability, overall health status, and control ability of the ANS. These findings demonstrate the positive effects of the SCG program on stress levels through proper physical activities in the community garden and various communication and exchange activities within and outside the community.

In contrast, the HEG reduced TP, sympathetic nerve activity, and stress resistance (Table 8). Specifically, it showed lower levels of TP, LF/HF ratio, and stress resistance than the SCG and NPG (Table 7), which suggests that the community garden program focusing on horticultural education increased the participants’ stress levels. This result counters previous findings that autonomous workplace gardening activities reduce the mean stress level of public officials [36] and that high school girls’ mean stress level is reduced after participating in gardening activities [37]. Gaining horticulture-related knowledge and skills or focusing on applying them for successful cultivation may increase anxiety and worry, and failure may result in energy depletion or resilience degradation. An approach that considers interventions related to psychosocial functions empowers the participants to cope with such failures or conflicts with neighbors by providing them with the mental strength to overcome them. Comparing these two approaches, it can be concluded that if the program focus is solely on acquiring knowledge and skills, participants are susceptible to stressors due to the lack of intervention in conflicts that may occur during the program implementation. Therefore, when offering a community garden program focusing on horticultural education, care should be taken to provide the participants with qualitative content aimed at reducing stress and increasing ANS activity.

### 4.3. Limitations

The main limitations are poor generalizability as the sample size of each experimental group was insufficient, the programs were implemented at different apartment complexes, and the participants were not randomly selected. To overcome these limitations, follow-up studies should be repeatedly conducted with homogeneous samples recruited from the same apartment complex and environment, as well as with programs providing interventions related to properly controlled content by program facilitators.

Despite its limitations, which have been presented in the research method section, this study is significant as it analyzed and compared two experimental groups and one control group using two different interventional approaches, instead of using one experimental group and one control group.

## 5. Conclusions

This study was conducted to test the effects of apartment community garden programs on the sense of community and stress by performing experiments in three groups: (1) SCG: the group participating in the program based on the sense of community theory (*n* = 11); (2) HEG: the group participating in the program with a focus on horticultural education (*n* = 11); and (3) NGP: the non-participation group (*n* = 10). The experiments led to the finding that the SCG program had a positive effect on the sense of community, TP, LF/HF, and stress resistance; the HEG program had a negative effect on TP, sympathetic nerve activity, and stress resistance.

From these results, it can be concluded that an apartment community garden program with appropriate intervention aimed at fostering a sense of community can have a positive impact on stress levels, whereas one that focuses on horticultural education without an appropriate intervention can decrease the sense of community and increase the stress levels of participants. 

The practical implications of this study highlight the importance of providing a customized program through intervention to foster a sense of community when operating an apartment community garden program and the role of urban gardening specialists who can efficiently provide guidance and encouragement. Such an apartment community garden will help boost the residents’ sense of community and reduce their stress.

## Figures and Tables

**Figure 1 ijerph-19-00708-f001:**
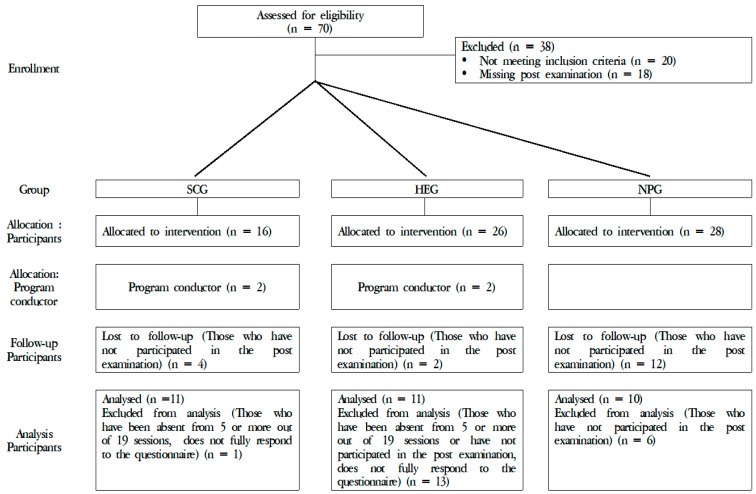
CONSORT (Consolidated Standards of Reporting Trials) flow diagram for individual controlled trials.

**Figure 2 ijerph-19-00708-f002:**
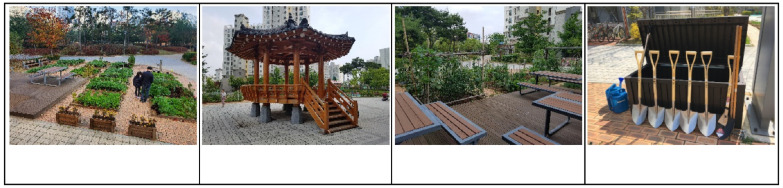
Site of the SCG program. SCG: group participating in the apartment community garden program based on the sense of community theory.

**Figure 3 ijerph-19-00708-f003:**
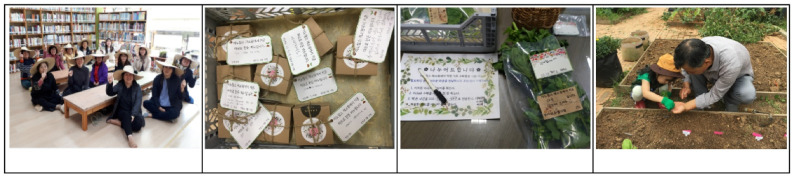
Snapshots showing the process of the SCG program. SCG: group participating in the apartment community garden program based on the sense of community theory.

**Figure 4 ijerph-19-00708-f004:**
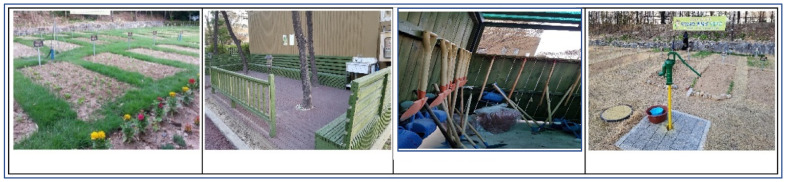
Site of the HEG program. HEG: group participating in the apartment community garden program with a focus on horticultural education.

**Figure 5 ijerph-19-00708-f005:**
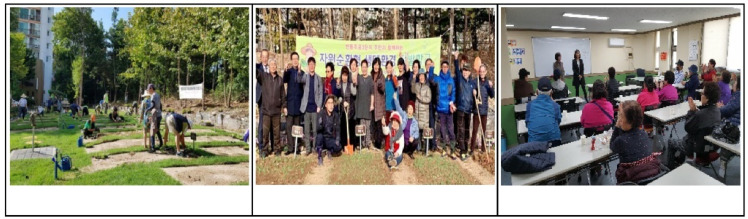
Snapshots showing the process of the HEG program. HEG: group participating in the apartment community garden program with a focus on horticultural education.

**Table 1 ijerph-19-00708-t001:** The research design for this study.

Group	Research Process		
SCG	O_1_	X	O_2_
HEG	O_1_	Y	O_2_
NPG	O_1_		O_2_

SCG: group participating in the apartment community garden program based on the sense of community theory; HEG: group participating in the apartment community garden program with a focus on horticultural education; NPG: non-participation group; O_1_: Pre-test, O_2_: Post-test; X: Apartment community garden program based on the sense of community theory; Y: Apartment community garden program focusing on horticultural education.

**Table 2 ijerph-19-00708-t002:** Participants’ age, sex, and the test of homogeneity.

Category		SCG (*n* = 11)	HEG (*n* = 11)	NPG (*n* = 10)	F	*p*
Age		59.2 ± 13.2 ^z^	66.6 ± 8.1	56.6 ± 5.8	2.893 ^y^	0.072 ^NS^
Sex	Male	2 (6.3) ^y^	0 (0)	2 (6.3)	2.410 ^w^	0.300 ^ns^
	Female	9 (28.1) ^x^	11 (34.4)	8 (25)		

SCG: group participating in the apartment community garden program based on the sense of community theory; HEG: group participating in the apartment community garden program with a focus on horticultural education; NPG: non-participation group; ^z^: Values are mean ± standard deviation; ^y^: Values are *F* by ANOVA; ^x^: Values are frequency(percent); ^w^: Values are χ^2^ by the Chi-square test; ^NS^: Non-significant at *p* < 0.05 leveled by ANOVA; ^ns^: Non-significant at *p* < 0.05 leveled by Chi-square test.

**Table 3 ijerph-19-00708-t003:** Application of the sense of community theory of McMillan and Chavis [18] to the apartment community garden program.

Components of a Sense of Community		Application Examples in the Apartment Community Garden Program
Membership	Common symbol system	Uniform caps, gloves, and scarves
	Boundaries	Elements: apartment garden and uniform name tags Interest: community garden program and horticultural knowledge Others: common symbol system
	Emotional safety	Emotional safety due to a common symbol system and clear boundaries
	Personal investment	Sharing baskets, maintaining garden tool storage boxes, and apartment flower bed care
	Sense of belonging and identification	Emotional safety and personal investment leading to a sense of belonging and identification
Influence		Within community: rule-setting, team formation, and role distribution Expansion of community: sharing basket, message cards, and community garden activities with daycare center children
Integration and fulfillment of needs		Integration of needs: pre-survey Fulfillment of needs: participants’ opinion convergence, horticultural knowledge, harvest utilization information, and provision of resources (compost, seeds, seedlings, agricultural tools, etc.)
Shared emotional connection		Space sharing: apartment complex and apartment community garden Time-sharing: time shared through participation in the program Experience sharing: joint efforts to organize shared activities, such as picnics and field trips, and overcome community garden management crises, such as pest infestation

**Table 4 ijerph-19-00708-t004:** The SCG program.

Session	Date (M/D)	Activities	Intervention for a Sense of Community
1	3/11	Orientation	Integration of needs
2	3/18	Uniform cap production and rule-setting	Membership (common symbol system) and influence (rules)
3	4/1	Gardening tool preparation, role distribution, and soil preparation	Integration and fulfillment of needs and influence
4	4/15	Harmonious planting plan	Integration and fulfillment of needs and membership (sense of belonging and identification)
5	4/29	Understanding companion plants	Influence
6	5/13	Gardening crisis (pest management)	Integration and fulfillment of needs and shared emotional connection
7	5/27	Building earthworm compost boxes and brunch making and sharing	Membership (personal investment) and shared emotional connection
8	6/10	Herbal soap making and sharing	Integration and fulfillment of needs and influence
9	6/24	Flower arrangement and sharing	Integration and fulfillment of needs and influence
10	7/8	Making plates and bookmarks decorated with pressed flowers	Membership (personal investment) and shared emotional connection
11	8/5	Making tomato and plum pickles	Membership (personal investment) and shared emotional connection
12	8/19	Sowing autumn seeds	Integration and fulfillment of needs
13	9/2	Sowing autumn plants with the children of the daycare centers in the complex	Membership (personal investment) and influence
14	9/16	Making marigold tea	Integration, fulfillment of needs, and membership (personal investment)
15	9/30	Transplanting Korean cabbage seedlings	Integration and fulfillment of needs
16	10/14	Field trip to gardening facilities	Shared emotional connection (reinforcement)
17	10/28	Dyeing handkerchiefs with marigold	Membership (common symbol system) and fulfillment of needs
18	11/11	Planting daffodil bulbs in the apartment community flower bed	Fulfillment of needs and shared emotional connection (expansion)
19	11/25	Garden party	Influence

**Table 5 ijerph-19-00708-t005:** The HEG program.

Session	Date (M/D)	Activities
1	03/29	Orientation
2	04/12	Setting rules, voting for a leader, creating a common garden, and planting leafy vegetables
3	04/19	Horticultural education focusing on nutritional management and planting flowers in the flower beds around the community garden
4	04/26	Understanding the utilization and characteristics of herb cultivation and planting flowers in the flower beds around the community garden
5	05/03	Horticultural education focused on pest management and planting seedlings of fruit vegetables (pepper and eggplant)
6	05/17	Soil education and prop construction
7	05/31	Planting succulent plants
8	06/14	Small-unit compost packaging
9	06/28	Potato harvesting
10	07/12	Making a plate garden
11	07/26	Pest control and field cleanup
12	08/09	Community field preparation for cultivating autumn plants
13	08/16	Kimchi vegetable planting and individual plant management
14	08/30	Handkerchief dyeing with marigold
15	09/06	Field trip to the forest on the outskirts
16	09/20	Garden pest management and organic liquid fertilizer preparation
17	10/04	Understanding herbal farming characteristics and herbal beverage preparation
18	10/25	Autumn plants cultivation education and compost application for autumn plants
19	11/08	Flower-pot production

**Table 6 ijerph-19-00708-t006:** Pre-intervention homogeneity test of the sense of community and stress among the three groups.

Group Item		SCG (*n* = 11)	HEG (*n* = 11)	NPG (*n* = 10)	*p*
Sense of Community (overall)		74.10 ± 10.67 ^z^	76.60 ± 8.42	67.10 ± 10.39	0.048 *
Membership		19.90 ± 2.77	20.22 ± 3.10	16.20 ± 2.97	0.413
Influence		13.10 ± 4.84	12.60 ± 3.35	14.10 ± 1.85	0.652
Fulfillment of needs		21.00 ± 3.09	22.78 ± 3.35	19.00 ± 3.37	0.183
Emotional connection		20.10 ± 2.81	21.00 ± 3.81	17.80 ± 3.19	0.563
Perceived stress		17.00 ± 7.85	24.86 ± 13.37	29.30 ± 5.23	0.005 *
Measured stress (biomarkers)	TP (Ln)	7.26 ± 0.61	7.43 ± 0.80	7.64 ± 0.79	0.388
	LF (Ln)	5.65 ± 1.02	5.54 ± 0.99	5.99 ± 0.89	0.517
	HF (Ln)	4.92 ± 0.78	4.92 ± 0.66	6.04 ± 0.83	0.008 **
	LF/HF (Ln)	1.15 ± 0.13	1.12 ± 0.13	0.99 ± 0.07	0.011 **
	SDNN (ms)	24.19 ± 9.69	25.35 ± 15.10	35.26 ± 10.85	0.077
	RMSSD (sqrt [ms])	17.41 ± 5.78	17.35 ± 8.55	36.10 ± 14.83	0.005 **
	Mean BPM (bpm)	78.08 ± 10.08	81.34 ± 15.42	72.40 ± 6.18	0.211

SCG: group participating in the apartment community garden program based on the sense of community theory; HEG: group participating in the apartment community garden program with a focus on horticultural education; NPG: non-participation group; TP: Total Power; LF: Low Frequency; HF: High Frequency; LF/HF: LF/HF ratio; SDNN: Standard Deviation of the NN interval; RMSSD: Root Mean Square of Standard Deviation; Mean BPM: Mean Beats Per Minute; ^z^: Values are mean ± standard deviation; *, **, ***: Significant at *p* < 0.05, 0.01, 0.001 leveled by the Kruskal–Wallis test.

**Table 7 ijerph-19-00708-t007:** Post-intervention comparison of the sense of community and stress among the three groups.

Group Item		SCG (*n* = 11)	HEG (*n* = 11)	NPG (*n* = 10)	F	*p*
Sense of community (overall)		^z^ 79.09 ± 3.02 b ^y^	65.51 ± 3.27 a	72.18 ± 2.67 a	4.862	0.015 *
Membership		^x^ 13.90 ± 2.33	11.89 ± 4.37	12.00 ± 4.51		0.054 ^ns^
Influence		^x^ 15.30 ± 1.70	11.44 ± 4.88	14.43 ± 2.50		0.087 ^ns^
Fulfillment of needs		^x^ 22.80 ± 2.39	18.56 ± 5.00	19.50 ± 3.78		0.051 ^ns^
Emotional connection		^x^ 18.10 ± 1.85 b	16.44 ± 2.51 a	16.00 ± 2.99 a		0.016 ^★^
Perceived stress		^z^ 19.98 ± 1.82 ^y^	23.15 ± 1.96	26.8 ± 1.77	3.135	0.063 ^NS^
Measured stress (biomarkers)	TP (Ln)	^x^ 7.69 ± 0.78 a	6.57 ± 0.79 b	7.42 ± 0.70 a		0.003 ^★★^
	LF (Ln)	^x^ 6.35 ± 0.85 a	4.82 ± 1.17 b	5.94 ± 0.67 a		0.002 ^★★^
	HF (Ln)	^z^ 5.80 ± 0.26 a	4.60 ± 0.26 b	5.34 ± 0.30 ab	5.978	0.007 **
	LF/HF (Ln)	^z^ 1.10 ± 0.05	1.08 ± 0.04	1.12 ± 0.05	0.135	0.874 ^NS^
	SDNN (ms)	^x^ 29.02 ± 13.31 a	15.65 ± 11.71 b	30.85 ± 10.13 a		0.001 ^★★^
	RMSSD (sqrt [ms])	^z^ 29.82 ± 4.22	18.98 ± 4.23	20.55 ± 5.19	2.016	0.152 ^NS^
	Mean BPM (bpm)	^x^ 78.73 ± 14.78 ab	86.30 ± 10.69 ab	72.15 ± 6.33 a		0.015 *

SCG: group participating in the apartment community garden program based on the sense of community theory; HEG: group participating in the apartment community garden program with a focus on horticultural education; NPG: non-participation group; TP: Total Power; LF: Low Frequency; HF: High Frequency; LF/HF: LF/HF ratio; SDNN: Standard Deviation of the NN interval; RMSSD: Root Mean Square of Standard Deviation; Mean BPM: Mean Beats Per Minute; ^z^: Values are estimated mean ± estimated standard error by ANCOVA; ^y^: Different letters within the line are statistically different at *p* = 0.05 according to Bonferroni test; ^x^: Values are mean ± standard deviation; ^NS^,*, **: significant at *p* ≤ 0.05, 0.01 leveled by ANCOVA; ^ns^, ^★^,^★★^: non-significant, significant at *p* ≤ 0.05, 0.01 leveled by the Kruskal–Wallis test.

**Table 8 ijerph-19-00708-t008:** Comparison of within-group pre–post means for sense of community and stress.

Item		^z^ Group	Pre-Test	Post-Test	z	*p*
Sense of community		SCG	^y^ 74.10 ± 10.67	80.60 ± 5.42	−2.193	0.028 *
		HEG	76.60 ± 8.42	68.56 ± 11.99	−1.474	0.141
		NPG	67.10 ± 10.39	68.20 ± 15.02	−0.306	0.759
Membership		SCG	19.90 ± 2.77	20.80 ± 1.48	−1.144	0.253
		HEG	20.22 ± 3.10	20.56 ± 2.01	−1.703	0.089
		NPG	16.20 ± 2.97	16.50 ± 5.32	−0.297	0.766
Influence		SCG	13.10 ± 4.84	15.30 ± 1.70	−1.178	0.239
		HEG	12.60 ± 3.35	11.44 ± 4.88	−0.357	0.721
		NPG	14.10 ± 1.85	14.10 ± 2.77	−0.175	0.861
Fulfillment of needs		SCG	21.00 ± 3.09	22.80 ± 2.39	−1.590	0.112
		HEG	22.78 ± 3.35	18.56 ± 5.00	−1.540	0.123
		NPG	19.00 ± 3.37	19.30 ± 4.03	−0.180	0.857
Emotional connection		SCG	20.10 ± 2.81	21.70 ± 2.11	−1.867	0.062
		HEG	21.00 ± 3.81	18.00 ± 4.03	−1.130	0.258
		NPG	17.80 ± 3.19	18.30 ± 4.52	−0.493	0.622
Perceived stress		SCG	17.00 ± 7.85	16.30 ± 6.73	−0.409	0.683
		HEG	24.86 ± 13.37	23.86 ± 10.21	−0.423	0.672
		NPG	29.30 ± 5.23	30.00 ± 4.29	−0.416	0.678
Measured stress (biomarkers)	TP (Ln)	SCG	7.26 ± 0.61	7.69 ± 0.78	−1.971	0.049 *
		HEG	7.43 ± 0.80	6.57 ± 0.79	−2.404	0.016 *
		NPG	7.64 ± 0.79	7.42 ± 0.70	−1.307	0.191
	LF (Ln)	SCG	5.65 ± 1.02	6.35 ± 0.85	−2.224	0.026 *
		HEG	5.54 ± 0.99	4.82 ± 1.17	−2.179	0.029 *
		NPG	5.99 ± 0.89	5.94 ± 0.67	−0.140	0.889
	HF (Ln)	SCG	4.92 ± 0.78	5.58 ± 0.83	−2.136	0.033 *
		HEG	4.92 ± 0.66	4.38 ± 0.97	−1.913	0.056
		NPG	6.04 ± 0.83	5.82 ± 1.00	−0.307	0.759
	LF/HF (Ln)	SCG	1.15 ± 0.13	1.15 ± 0.09	−0.277	0.782
		HEG	1.12 ± 0.13	1.10 ± 0.19	−0.411	0.681
		NPG	0.99 ± 0.07	1.05 ± 0.18	−1.211	0.226
	SDNN (ms)	SCG	24.19 ± 9.69	29.02 ± 13.31	−1.913	0.056
		HEG	25.35 ± 15.10	15.65 ± 11.71	−2.223	0.026 *
		NPG	35.26 ± 10.85	30.85 ± 10.13	−1.376	0.169
	RMSSD (sqrt [ms])	SCG	17.41 ± 5.78	25.15 ± 15.59	−1.778	0.075
		HEG	17.35 ± 8.55	14.25 ± 11.83	−1.275	0.202
		NPG	36.10 ± 14.83	30.88 ± 18.31	−1.070	0.285
	Mean BPM (bpm)	SCG	78.08 ± 10.08	78.73 ± 14.78	−0.178	0.859
		HEG	81.34 ± 15.42	86.30 ± 10.69	−1.734	0.083
		NPG	72.40 ± 6.18	72.15 ± 6.33	−0.153	0.878

SCG: group participating in the apartment community garden program based on the sense of community theory; HEG: group participating in the apartment community garden program with a focus on horticultural education; NPG: non-participation group; ^z^: based on *n* = 11 for SCG, *n* = 11 for HEG, *n* = 10 for NPG; ^y^: Values are mean ± standard deviation; *: Significant at *p* ≤ 0.05 leveled by the Wilcoxon signed-rank test.

## Data Availability

Data will be available on request to the corresponding author.

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
