# Peer review of "Effect of Apartment Community Garden Program on Sense of Community and Stress"

_ijerph, 2022, doi:10.3390/ijerph19020708_

Round 1

Reviewer 1 Report

Overall good paper and good design but as I always make this comment, where is the lit. review on the issue of community conflict, sense of civicness and community gardens. 

Reviewer 2 Report

I think the topic is of interest. There are a number of items that need to be improved. 

Introduction must be improved

  • The authors assume “a customized horticultural program using an apartment community garden can help increase residents’ sense of community and reducing stress”. They state reasons for their assumption; however, they state that as a given.
  • It is confusing to have the apartment complexes for both experimental groups to be labeled “L”
  • There is no description of the “related program” which makes it seem that both groups has similar past experiences
  • The theoretical perspective that is described in the Materials section would be better located in the Introduction

Research Design needs to be clarified

  • The information in Table 1 is hard to follow. Would be better served with text than with just having letters.
  • Some of the citations do not follow the proper convention
  • The data analysis should be explained in one section. There are too many references to data analysis throughout the paper
  • The measurement tools require identification in the opening paragraph of the 2.4 section
  • The biomarkers need more explanations, including the rationale for their inclusion

Methods and Materials need to be better described for reader understanding

  • Materials would be better identified as Methods
  • SCG program and HEG program should be in a bulleted lists rather than dense paragraphs
  • References to Figure 2 and Figure 4 are several paragraphs above the actual figures, thus being confusing

 Results section needs to be improved

  • The data tables seem adequately done
  • The Discussion section would be better understood with references to the data tables
  • The Discussion section is very dense and difficult to follow

Conclusions needs to be expanded

  • The term “appropriate intervention” only appears here and is not described in terms of the elements of the experiment. The reader is left with the question; Are they referring to the SCG program?
  • Limitations should be in the Discussion not that Conclusions

In addition, some Moderate English changes required

  • The topic sentence in some paragraphs is hard to determine, therefore the information is not clear

Reviewer 3 Report

Dear authors

Thank you for the opportunity to review your article.

Brief summary: This is a quasi-experimental study that was conducted to investigate the effect of a community garden program in an apartment complex in fostering residents’ sense of community and reducing stress.

Areas of strength

The references included are relevant for the subject under study although it only presents 8/29(28%) references from the last 5 years. There is strong concordance between the and the methods used. The description of the methodology was made in a clear and adequate way, although the CONSORT flow diagram is missing. The results are clearly described. The discussion correlates with the presented data and takes the published literature into account. The manuscript presents some limitations and clinical implications.

Aspects to improve:

  1. increase the number of bibliographic references in the last 5 years.
  2. Pag 3, line 93 - Please enter the Flow diagram CONSORT (The manuscript has been written following CONSORT guidelines for non-pharmacological?)
  3. Page 9, line 277 - refer to data presentation. Continuous quantitative data were presented as mean and standard deviation and qualitative data as n and % [example Results are presented as the mean ± standard deviation or n(%)]
  4. Page 15-16. The references do not follow the journal guidelines, the journal name is missing and the year must be written in bold.

Round 2

Reviewer 2 Report

This article shows much improvement. Once the edits are accepted I think this article is ready to publish.

Author Response

Dear Dr Ruby Liu,

Thank you for giving me an opportunity to submit a revised draft of my manuscript titled Effect of Apartment Community Garden Program on Sense of Community and Stress to the International Journal of Environmental Research and Public Health.

We appreciate the time and effort that you and the reviewers have dedicated to providing your valuable feedback on my manuscript. We are grateful to the reviewers for their insightful comments on my paper. We have been more faithful in editing and proofreading.

Comments from Reviewer 2 

  • Comment 1: This article shows much improvement. Once the edits are accepted I think this article is ready to publish.

Response: All spelling and grammatical errors pointed out by the reviewers have been corrected.

Thank you very much for guiding us to improve our article.

Sincerely, 
